# The impact of emotional intelligence on operational effectiveness: The mediating role of organizational citizenship behavior and leadership

**Ricardo Santa** [1]*, **Andreina Moros**[1], **Diego Morante**[2], **Dorys Rodríguez**[1], **Annibal Scavarda**[3]

**1** Cesa–Colegio de Estudios Superiores de Administración, Bogotá, Colombia, **2** Emavi–Escuela Militar de Aviación, Cali, Colombia, **3** Federal University of the State of Rio de Janeiro, Rio de Janeiro, Brazil

* ricardo.santa@cesa.edu.co, rsantaaus1@gmail.com

**Data Availability Statement:** https://doi.org/10.57130/CESA.4947 Data used in the analysis is

## Abstract

### Purpose

This article examines the influence of emotional intelligence on organizational citizenship behavior and transformational and transactional leadership, and the impact of these dimensions on operational effectiveness.

### Design/Methodology

The analysis was based on 180 valid questionnaires from organizations in Colombia's manufacturing sector of the Valle del Cauca region. The variables were analyzed using structural equation modeling to identify the relationships among the studied constructs.

### Findings

The results suggest that emotional intelligence positively affects organizational citizenship behavior. Nevertheless, emotional intelligence does not impact transformational leadership and only partially affects transactional leadership and operational effectiveness. Emotional intelligence has a strong and positive impact on operational effectiveness when mediated by organizational citizenship behavior, which does have a strong and positive predictive power on operational effectiveness. Hence, in the search for competitive advantage, leaders should seek to improve operational effectiveness by focusing on developing emotional intelligence and organizational citizenship behaviour skills. Interestingly, of the two leadership styles examined in this study, only transactional leadership impacts operational effectiveness, which is inconsistent with the current literature and indicates a need for further leadership training.

### Originality/Value

The value of this paper lies in discerning the current capabilities and strategies that individuals in an organization must address for proper transactional and transformational

available without restrction at Cesa's Library repository.

**Funding:** The authors received no specific funding for this work.

**Competing interests:** The authors have declared that no competing interests exist.

leadership. However, before operational effectiveness and a sustainable competitive advantage can be achieved, the role of leaders should be managed through the appropriate application of the concepts of emotional intelligence and organizational leadership behavior.

## Introduction

Business strategy and operational effectiveness play a determining role in the competitiveness of manufacturing companies due to the changing and volatile context of the markets [1]. Today's organizations seek to operate better and faster than the competition, which implies that for a company to lead the market, it needs to make strategic decisions on how to carry out its processes [2]. These decisions about competitiveness bring to light the need to examine whether their employees have the leadership capabilities and skills necessary to achieve a better market share [3]. Although leadership significantly affects employee performance [4], few empirical studies show how leadership style and soft skills, such as emotional intelligence and organizational citizenship behaviour, affect operational effectiveness. Consequently, this research aims to understand the impact of emotional intelligence and organizational citizenship behavior on operational effectiveness when mediated by transformational and transactional leadership.

Organizations in the manufacturing sector seek operational effectiveness by reducing costs and improving the quality of their processes [5, 6]. Employees' skills and knowledge can be crucial assets for this purpose, regardless of the operational level of the organization's position in the market [7]. Studies in service sector organizations have empirically shown that emotional intelligence and organizational citizenship behavior significantly impact an organization's competitiveness [8]. Others have concluded that leadership style positively influences organizational performance and customer satisfaction [9]. Therefore, it is crucial to investigate the role of employees' soft skills in achieving sustainable competitive advantage. Additionally, it is essential to understand better the impact of emotional intelligence and organizational citizenship behavior on operational effectiveness when mediated by the leadership style exercised by employees.

This research aims to assess different dimensions of emotional intelligence, organizational citizenship behavior, and leadership styles to provide integrative empirical evidence of the impact of soft skills on operational effectiveness (Fig 1). In this context, emotional intelligence is recognizing, using, understanding, and managing emotions and emotional information [10]. Organizational citizenship behavior refers to voluntary and spontaneous behaviors that extend beyond the regular job duties of employees [11, 12].

This article is organized as follows. The second section provides the literature review supporting this study's model. The third section presents the hypothetical model and the methods of empirical analysis. The fourth section offers the quantitative results, and the fifth section discusses these results and provides the analysis and implications of the findings. The last section concludes the investigation.

## Literature review

### Emotional Intelligence (EI)

Emotional intelligence is rooted in the social intelligence research done by American psychologist Edward Thorndike in 1920. Emotional intelligence, according to Thorndike, is the

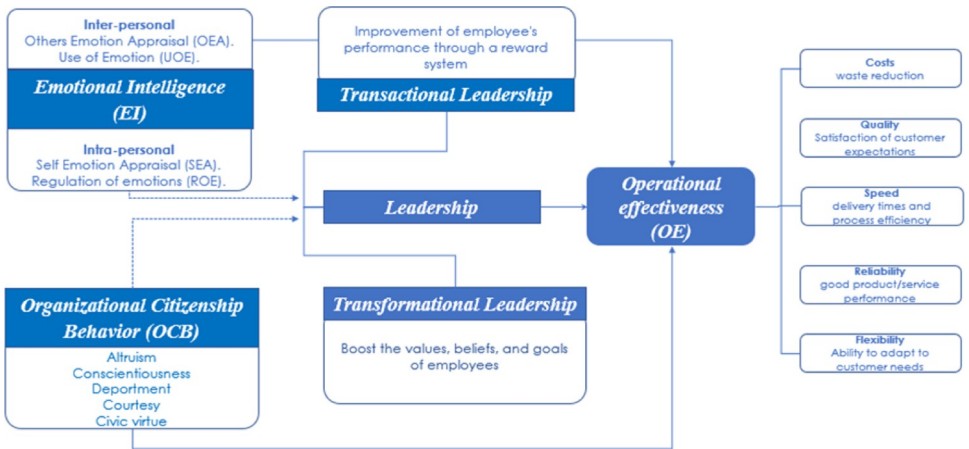

**Fig 1. Conceptual framework used to evaluate the impact of emotional intelligence and organizational citizenship behavior on leadership and operational effectiveness.** (Developed for this study by the authors).

capacity for people to comprehend others and behave wisely in social situations. This idea suggests that intelligence may be broken down into three categories: abstract, mechanical, and social intelligence [13].

Understanding relational intelligence and the theory of multiple intelligences became popular in the 1980s through the work of Gardner [14]. Interpersonal intelligence (the capacity to communicate with and understand others) and intrapersonal intelligence (the ability to understand oneself and use this knowledge to work effectively in life) are two of Gardner's eight categories of intelligence. Based on this theory, Salovey and Mayer [15] organized the idea of emotional intelligence, defining it as a collection of meta-skills that can be divided into five competencies: awareness of one's own emotions, capacity for emotional regulation, capacity for self-motivation, awareness of other people's emotions, and control over interpersonal interactions.

Goleman's contribution to popularizing the idea of emotional intelligence in the business world must also be acknowledged. In his 1995 book, he describes the application of emotional intelligence in management and the resulting range of advantages. He defines emotional intelligence as the capacity to reason effectively using emotional knowledge, which includes the capacity to perceive, regulate, and manage emotions. It is the ability to recognize one's feelings and those of others to motivate and control emotions and relate to others [10].

According to Salovey and Mayer [15], emotional intelligence is a type of relationship intelligence since it is a skill needed to guide human capital or employees in the right decision-making. There is a strong relationship between high emotional intelligence and adequate leadership that affects an organization's ability to induce organizational citizenship behavior. Salovey and Mayer validated the assessment of four factors that govern the development of emotional bonds between various parties and enable people to create fulfilling relationships with others:

1. Self Emotion Appraisal (SEA). The person can recognize, comprehend, and communicate his or her feelings. It is distinguished by aiding in the development and maintenance of interpersonal relationships. This idea supports the growth of judgment and decision-making.

2. Others' Emotion Appraisal (OEA). This refers to the person's ability to recognize and comprehend other people's emotions and to respond accordingly. Because they can understand

the views of others, they are capable of perceiving and absorbing their feelings as well as those of others.

3. Regulation of emotions (ROE). This entails having the ability to control one's own emotions. It suggests the ability for emotional restraint, which will allow the person to recover from psychological anguish, as well as the capacity for unanticipated changes in outcomes or circumstances, which will lead to flexibility and a wider range of problem-solving techniques.

4. Use of Emotion (UOE). People can harness their own emotions to enhance both individual and group performance. Thus, having a good attitude encourages imagination and creativity, whereas having a bad attitude invites a useless and endless informational process.

These dimensions have been used for recent studies. For example, Santa et al. [16] established a model with similar items and affirmed that the ability to perceive and understand the emotions of others increases an employee's commitment to being operationally effective. This commitment on the part of the employees would lead companies to differentiate themselves in the market and achieve operational effectiveness, which translates into a competitive advantage.

Emotional intelligence includes skills such as self-awareness, self-regulation, self-motivation, empathy, and social competence [17]. As a result, emotional intelligence cannot be seen as an aspect of a person's personality, but rather as a competency related to human behavior that helps to understand how others behave and which, based on this understanding, has the potential to be enhanced and developed [18].

Emotional intelligence and several of its dimensions have been linked to different leadership styles [19]. A leader with great emotional intelligence can recognize others' feelings and use this knowledge to persuade team members [20]. Research on organizational behavior, human resources, and management has shown that the level of an individual's emotional intelligence can predict his or her leadership, negotiation, interpersonal relations, work-home balance, and work performance capacities [21]. An individual whose emotional intelligence is considered high is an individual who can potentially become a leader within the organization capable of mobilizing teams and impacting to a certain extent the results of the operation [22]. Thus, emotional intelligence is essential for effective leadership, and self-awareness, self-management, and empathy are the most commonly used skills/competencies [23].

## Organizational Citizenship Behavior (OCB)

According to Khan et al. [24], organizational citizenship behavior is a requirement for effective organizations. Regular behaviors alone are not enough for organizations. Innovation and ongoing change depend on actions outside the scope of job descriptions. Such outstanding behavior is necessary for both survival and invention [25]. Organizational citizenship behavior can serve another purpose within organizations when used as a support system for collaboration in times of need or while seeking change. It can genuinely promote social interaction within a given setting [26]. By fostering teamwork and reducing conflict, organizational citizenship behavior improves organizational performance and production [27, 28].

Organizational citizenship behavior represents an individual´s discretionary behavior, not directly or explicitly recognized by the formal reward system, and in an aggregate way promotes the efficient and effective functioning of the organization. Organizational citizenship behavior is classified into five dimensions [11, 29].

1. Altruism refers to any discretionary behavior of employees in the form of helping other members of the organization in their specific tasks or relevant organizational problems, which generates a feeling that coworkers help each other.

2. Conscientiousness. Voluntary behaviors of the members of the organization that go beyond the minimal job requirements in certain aspects related to internal order such as attendance at work, punctuality, protection of resources, and being meticulous, honest, and careful, among others.

3. Deportment is the ability to tolerate, resist, and prevent the dejection resulting from the inevitable difficulties derived from work.

4. Courtesy refers to the continuous interaction between the members of the organization, who work for the shared purposes of the company, and to collective and positive behaviors, such as communicating to other members the work done and the decisions taken.

5. Civic virtue implies the development and support of political organizations and the participation of the entity's members in operations.

Methot et al. [30] categorize organizational citizenship behavior first into behaviors that are displayed toward specific members of the organization, such as being courteous and charitable. Second, they look at acts that are meant to help the organization, which includes things like sportsmanship. Love and Kim [31] and Methot et al. [30] described organizational citizenship behavior as an employee's conduct that is reflected in their devotion and loyalty to the organization in which they work, and for which they do not anticipate any compensation in exchange for their actions and behaviors to further the organization's goals.

We can conclude that these behaviors are not part of the tasks to be performed in their positions and therefore are not recognized by the formal reward system. However, employees develop constructive behaviors that are reflected in the performance and effectiveness of the organization, thus the first hypothesis of this study states:

Hypothesis 1 (H1): Employees' emotional intelligence (EI) positively impacts organizational citizenship behavior (OCB).

## Leadership

The main goal of human resources in a business is to effectively manage its workforce by encouraging positive employee attitudes like greater productivity, job satisfaction, and enthusiasm while decreasing unfavorable attitudes like higher turnover, tardiness, and disruptive behavior at work. For this reason, it is crucial to prioritize employee perceptions of the organization's leadership, processes, and policies as enablers of innovative outputs, because these perceptions can either promote or stifle creativity and innovation inside the organization [32].

Leadership is the activity or process of persuading others to willingly devote themselves to achieving the group's goals, with the group understanding a sector of the organization with shared interests [33, 34]. Leadership within a group of people and organizations will always be vital to the organization's effectiveness as it makes things happen [35, 36].

Two leadership styles and an inherent dissonance emerge from the literature: transactional leadership and transformational leadership, although uniquely different, have traits that can present and coexist simultaneously in the same leader or be complementary. Recent studies reflect different positions: Podsakoff et al. [38] opined that the transactional leadership style does not affect performance, but Purwanto et al. [83] showed in their study that both

transactional and transformational leadership styles have positive and significant solid effects on performance. The characteristics of each type of leadership are summarized below.

**Transactional leadership.** In this leadership style, the team leader must act as a change agent and achieve a significant transformation in his/her subordinates to improve productivity. This type of leadership is essentially a reward system, which is positively correlated with the subordinate's performance behavior [37]. Three factors can be used to analyze the interaction between transactional leaders and their subordinates: first, the leader will indicate what the subordinates will receive if their job is in line with expectations because the leader knows what the employees want. Second, the leader rewards the work that is put in by subordinates, and, third, leaders are receptive to the personal interests of their followers if such interests are equal to the importance of the tasks the followers perform [37].

**Transformational leadership.** This leadership style consists of the achievement of the goals set up by the organization's leader to obtain the organization's results without abandoning the team members' goals. Transformational leaders can motivate their followers to perform above expectations by providing them with goals and aspirations, articulating these in a vision and a suitable role model whose behavior is centered on contingent reward and punishment behavior [38].

Transformational leaders are seen as captivating people who inspire and encourage their team members instead of commanding them, engaging their intellect and emotional response, and thereby igniting commitment to the vision and mission of the organization. A transformational leader's ability to foster an environment where followers achieve above and beyond expectations is one quality [39]. In addition, they have experience, knowledge, and mastery over the activity they lead, they are energetic and do not avoid taking risks, they constantly challenge employees to think for themselves, and they raise the morale and motivation of the work team. Although this is a very effective leadership style, it has been shown that sometimes it must be combined with other leadership styles to ensure the efficiency of the organization's processes [40].

The transformational leadership style will result in an organization performing better and meeting the expectations for innovation and change coming from both the internal and external environment. Individual initiatives that can help the organization and changes brought about by the team members' creative practices are valued by the transformational leader, which is essential for the development of innovative practices [6].

Shahhosseini et al. [41] examined the relationships between emotional intelligence and leadership styles among 192 managers as part of recent research on these topics. The findings revealed that emotional intelligence was positively correlated with emotion in job performance and that the transformational leadership style was correlated with job performance. The findings imply that emotional intelligence might offer a novel and intriguing strategy to improve productivity through job performance. To better understand the relationships between emotional intelligence and leadership in a non-Western setting, 230 supervisors and subordinates from banking branches in the Indian state of Jammu and Kashmir were studied by Lone and Lone [42]. They emphasized the need to create strong tools for selecting, training, and developing leaders and revealed that emotional competency and emotional sensitivity were predictors of effective leadership. This might potentially improve organizational atmosphere and performance.

Mysirlaki and Paraskeva [43] investigated the effects of leaders' emotional intelligence and transformational leadership on virtual team effectiveness by taking a deeper look at the virtual world teams of 500 participants of massively multiplayer online games. This analysis included three sub-factors of team effectiveness: performance, viability, and member satisfaction. They revealed a significant predictive relationship between perceived leader emotional intelligence

and virtual team effectiveness sub-factors mediated by transformational leadership behavior. Thus, our study proposes:

Hypothesis 2 (H2): Employees' emotional intelligence positively impacts transactional leadership.

Hypothesis 3 (H3): Employees' emotional intelligence positively impacts transformational leadership.

However, there are no studies that analyze the impact of employee emotional intelligence on operational effectiveness, so we hypothesize the following:

Hypothesis 4 (H4): Employees' emotional intelligence positively impacts operational effectiveness.

## Leadership, organizational citizenship behavior, and emotional intelligence

According to Bass [44], the subordinates of transformational leaders spend more extended hours and produce more than is expected of them. When their subordinates need assistance, transformational leaders guide them, help them develop their skills, educate them, and treat each person equally [45]. It is a style of leadership that, according to its description, "manages organizations around a purpose in ways that inspire and advance the aspirations of employees" [46]. As a result, transformational leaders work to elevate their people's ambitions and standards while enhancing their knowledge and skills. As a result, the colleagues modify their objectives and ideals and grow closer together. According to earlier research, employee performance and business success are related [32, 47].

This kind of leadership is one of the predictors of organizational citizenship behavior [48]. Furthermore, transformational leadership may increase employees' organizational citizenship behavior [49]. The relationship between leadership, emotional intelligence, and organizational citizenship behavior has been carried out in previous research. Majeed and Jamshed [50] analyzed the direct and indirect effects of transformational leadership by examining the mediating role of emotional intelligence. The results from 220 responses indicated that the relationship between transformational leadership and organizational citizenship behavior is statistically significant.

Transactional leadership is described as a cost-benefit trade between leaders and followers [51]. The exchange or transaction involves anything of value between the follower's desired reward for providing services and the leader's possessions or control [52].

Because emotional intelligence entails the capacities for correct emotion perception, access and production of emotions to support thought, comprehension of emotions and emotional knowledge, and reflective regulation of emotions to support emotional and intellectual development [47], it plays an essential role as a mediator. It adds to the positive effects of the transformational leadership style interconnected with other role behaviors at work, making it more effective.

Abdullahi et al. [53] studied the role of leaders' emotional intelligence in moderating the relationship between leadership styles and organizational citizenship behavior using data from 618 employees of small and medium-sized enterprises (SMEs) in Ghana. The findings show that both democratic and transformational leadership styles positively predicted the organizational citizenship behavior of SME employees, with transformational leadership having a more significant influence.

Majeed and Jamshed [50] explored how transformational leadership influences organizational citizenship behavior by addressing the mediating role of workplace spirituality and

emotional intelligence. They used data from 408 academicians working in public sector universities. The findings provide empirical evidence and encouraging justification for the substantial influence of workplace spirituality and emotional intelligence on the relationship between transformational leadership and citizenship behaviors.

Likewise, D'Souza et al. [54] investigated the relationship between emotional exhaustion and performance during the COVID-19 pandemic with 384 respondents from the faculty and administrative personnel in the Mangalore Diocese educational institutions. The findings showed that organizational citizenship behavior moderated the link between performance and emotional weariness. Emotional intelligence and transformational leadership work together to affect organizational citizenship behavior to lessen the dysfunctional effects of emotional exhaustion. Thus, our study proposes analyzing the relationship between organizational citizenship behavior and leadership styles. Thus, we propose that:

Hypothesis 5 (H5): Organizational citizenship behavior positively impacts transformational leadership.

Hypothesis 6 (H6): Organizational citizenship behavior positively impacts transactional leadership.

## Operational Effectiveness (OE)

According to Porter [55], operational effectiveness is the capacity to outperform competitors at the same tasks by utilizing procedures based on the organization's core competencies. Therefore, outstanding performance is difficult for firms in a globalized economy with fierce competition and quick change. As a result, businesses seek ways to make processes and services more profitable and efficient [56].

Operational effectiveness is part of cost leadership, but it also applies to quality leadership and is one of the main drivers of business success. To achieve this goal, organizations focus their activities on attaining elevated levels of operational effectiveness, and it is necessary to analyze the following dimensions [5, 6, 57]:

Cost: All organizational components, including purchasing, innovation, development, and product design, among others, are included in cost performance. An organization can identify and eliminate inefficiencies and waste in service or production processes without squandering time, resources, or cash on tasks like staff performance, product or service design, or purchasing. Every company facet, including product development, innovation, and procurement, is involved in cost performance [58].

Quality: Customers have specific requirements for goods or services. That entails providing top-notch goods or services and attending to their demands. Giving clients what they want and how they want it is what quality is all about. However, it makes no difference whether the company provides services or products because quality requires various operating procedures. Production times, response times, wait times, delivery times, delays, warranty times, service personnel, service techniques, after-sales services, repair quality, locations, and accountability attitudes can all contribute to quality [59].

Reliability refers to meeting the customer's requirements for delivery timing, ensuring that the products or services are consistent, and ensuring that the organization's operations are focused on achieving that objective [60]. This indicates that the items function as intended during a specific time frame and certain environmental circumstances. Reliability is crucial for operational effectiveness since it directly affects customer satisfaction [61].

Flexibility refers to a company's ability to quickly set up various processes or routines in response to changes in client demands or market conditions. Organizations must build flexible

operations to survive increasingly competitive marketplaces and establish a competitive advantage [62, 63].

Speed can be characterized as the time needed to produce new products or services and to respond to customer demands. Speed is related to critical characteristics such as rapid adaptation, quick motions, and close relationships within all parts of an organization. Speed can be viewed as a crucial skill that every organization needs because of the frequent changes in the organizational environment [64].

Therefore, operational effectiveness entails enhancing and monitoring the efficiency of the organization's processes and managing and controlling the business's operations to boost its competitiveness [2]. Additionally, if a company performs better, faster, and more smoothly than its rivals, operational effectiveness may be the key to outperforming them in terms of results [65]. Measurement issues can be used to categorize the fundamental difficulties. Benefits, both quantitative and qualitative, are challenging to measure yet are often seen in service operating environments [66].

Consequently, to analyze the relationship between organizational citizenship behavior (OCB) and operational effectiveness (OE) and between each leader style and operational effectiveness (OE); therefore, we propose the following hypotheses:

Hypothesis 7 (H7): Organizational citizenship behavior positively impacts operational effectiveness.

Hypothesis 8 (H8): Transformational leadership positively impacts operational effectiveness.

Hypothesis 9 (H9): Transactional leadership positively impacts operational effectiveness.

The literature linking the constructs addressed in this study, especially in Colombia, is scarce. There is no evidence of studies investigating the influence of emotional intelligence on different leadership styles and, in turn, on organizational citizenship behavior and operational effectiveness. Therefore, after the review of the literature, we propose the following hypothetical model depicted in Fig 2.

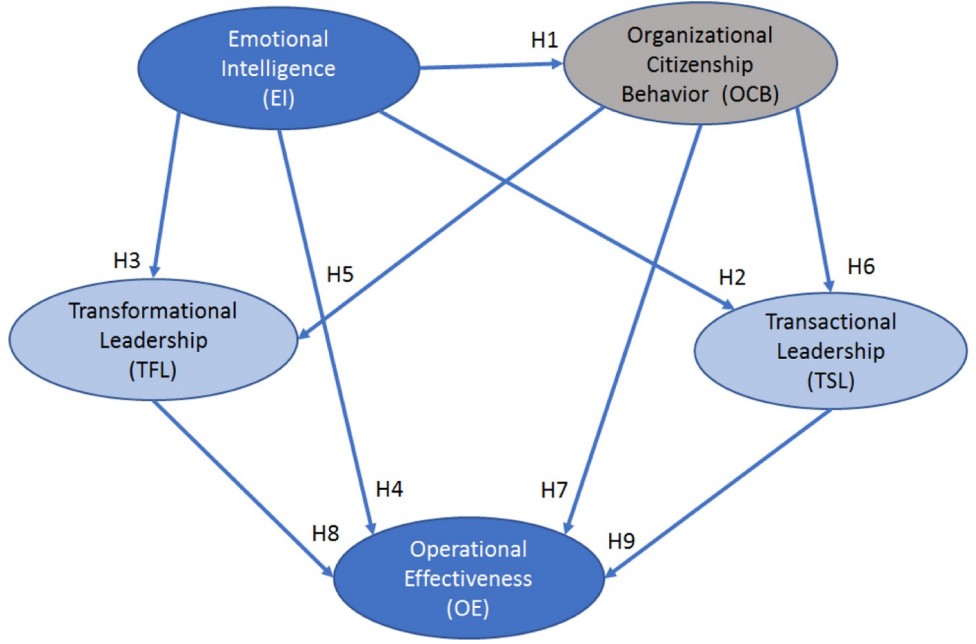

**Fig 2. The hypothetical model developed for this study.**

## Research methods

In this study, the hypotheses shown in Fig 2 were explored using a survey instrument created following the suggestions offered by Hair et al. [67] in their best-fit model. Recognizing the benefits of online surveys [57], we distributed invitations to prospective participants that included a link to the survey in Microsoft Forms, from which the data can be converted to SPSS.

We carefully chose responders from a convenience sample, considering each person's position, experience, knowledge, and tenure. The studied population consisted of 180 respondents affiliated with organizations in the manufacturing sector in Colombia's Valle del Cauca industrial region. We selected managers, engineers, and employees with leadership responsibilities. Each received questionnaire was reviewed for completeness; several were considered unusable due to inconsistencies or significant missing data, or lack of involvement of the respondent in leadership responsibilities. In total, 76 questionnaires were discarded. The response rate was 59.30%.

The indications of Hair et al. [67] were followed to support the sample size used in this research. The minimum sample for an SEM model must be larger than the number of covariances in the data matrix input. Additionally, the minimum appropriate sample is made up of 10 respondents per parameter. Therefore 180 valid responses are more than sufficient for an SEM model with five constructs.

The questionnaire was based on an extensive literature review, and four sections were designed. Statements related to the operationalization of the various constructs were evaluated based on a five-point Likert-type scale (Strongly Disagree—Strongly Agree) representing the respondents' perceptions according to the different questions. The first part of the questionnaire was designed to profile the participants. The second section included 16 statements about emotional intelligence based on the scale developed by Wong and Law [68].

The Leadership Style Questionnaire [CELID] [69] was used to quantify the transformational and transactional leadership variables. Based on Bass's leadership theory, this questionnaire strictly specifies these variables in measurable elements [70]. The Multifactor Leadership Questionnaire, sometimes known as the MLQ, is the source of the CELID instrument. It provides insight into the three most common leadership philosophies: transformational, transactional, and laissez-faire (lack of leadership) [70]. Additionally, it includes two surveys with 34 self-administered response options for leaders (CELID-A) and subordinates (CELID-S).

The third section was elaborated following the works of Asgari et al. [9] to measure the behavioral variables of organizational citizenship. Finally, an adaptation of the questionnaire developed by Santa et al. [2] and Santa et al. [71] was used to analyze the construct related to operational effectiveness. S1 Appendix shows the measurement items.

To avoid the social desirability bias, the ethics committee at the Colegio de Estudios Superiores de Administración (CESA) in Colombia reviewed the questionnaire to ensure that questions were neutral, unbiased, and non-threatening. Additionally, before filling out the questionnaire, the respondents were informed that all information provided would be treated in the strictest confidence, that the responses would be aggregated and used for research purposes, that by completing this survey, they were giving their consent to such use, and that they might withdraw at any time. The researchers followed ethical procedures during this research project by adhering to ethical standards.

**Data analysis.** The structural equation model's variables were built by using the average mean values of the statements' ratings. This methodology fits the appropriateness for our research, the analysis of latent variables and their cross relationships, and the fitness between the required sample and the data collected [72]. We used SPSS and AMOS software version 28

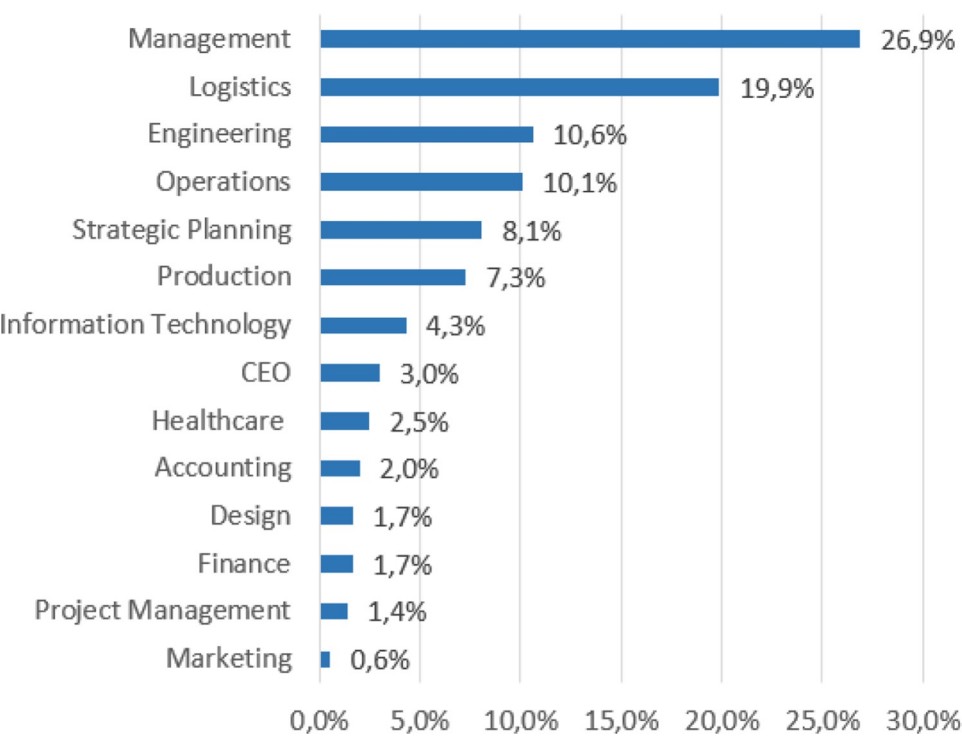

**Fig 3. Respondents per area of responsibility.**

to analyze the data. The analyses included confirmation of the conceptualized model shown in Fig 2.

Fig 3 shows the demographics for this study by the respondent's area of responsibility. The data collected shows that 27% of the respondents were involved in managerial responsibilities, 20% in logistics, and 10% in engineering and operations. Another important factor about the sample is that many respondents have jobs related to operational activities.

Confirmatory factor analysis (CFA) was used to analyze the links between continuous latent and observable variables and estimate how well the model fits the data overall [67, 73]. Factor loadings were initially calculated, and the items were only loaded on one construct. Internal consistency was evaluated using the items-to-total correlation and Cronbach's alpha coefficient to correlate the latent constructs. The coefficient values for the constructs are listed in Table 1. The statistical analysis carried out to calculate the model's predictiveness and its indices are described in the following steps.

1. Confirmatory factor analysis (CFA) with the following settings:

2. Discrepancy: Asymptotically distribution-free

3. Covariances supplied as input: Unbiased

4. Covariances to be analyzed: Maximum likelihood

5. Factor loading estimation. (Assessment of the relationship between the observed and continuous latent variables.)

**Table 1. Reliability measures.**

| | | Factor Loading | Cronbach's Alpha | Composite Reliability CR | Average Variance Extracted AVE |
|---|---|---|---|---|---|
| Emotional Intelligence—EI | EI1 | 0.750 | 0.961 | 0.912 | 0.599 |
| | EI2 | 0.790 | | | |
| | EI3 | 0.640 | | | |
| | EI4 | 0.770 | | | |
| | EI5 | 0.820 | | | |
| | EI6 | 0.810 | | | |
| | EI7 | 0.820 | | | |
| Organizational Citizenship Behavior—OCB | OCB1 | 0.770 | 0.902 | 0.784 | 0.549 |
| | OCB2 | 0.750 | | | |
| | OCB3 | 0.770 | | | |
| Transformational Leadership—TFL | TFL1 | 0.720 | 0.951 | 0.881 | 0.651 |
| | TFL2 | 0.820 | | | |
| | TFL3 | 0.870 | | | |
| | TFL4 | 0.810 | | | |
| Transactional Leadership—TSL | TSL1 | 0.770 | 0.849 | 0.812 | 0.520 |
| | TSL2 | 0.750 | | | |
| | TSL3 | 0.690 | | | |
| | TSL4 | 0.670 | | | |
| Operational Effectiveness—OE | OE1 | 0.790 | 0.965 | 0.853 | 0.539 |
| | OE2 | 0.770 | | | |
| | OE3 | 0.740 | | | |
| | OE4 | 0.730 | | | |
| | OE5 | 0.630 | | | |

6. Internal consistency. All Items above 0.70 [67]. Cronbach's alpha coefficient is shown in Table 1.

7. Testing of construct validity (CFA).

The state of multicollinearity among dimensions was examined using AMOS. There was no evidence of multicollinearity in the data. The variance inflation factor (VIF), derived from the regression analysis, was used to determine how much one independent variable interacted with another. A typical assessment of multicollinearity is the VIF value.

To avoid overfitting, we considered the fitting propensity of the model through the evaluation of the parsimony indices, PNFI = 0.755, PCFI = 0.832, and PGFI = 0.695 [74].

We compared the square root of the average variance extracted (AVE) to test the model's reliability. AVE measures the amount of indicator variance explained by the latent variable relative to the amount due to measurement error [75]. Ideally, the AVE should be >0.50. This means that the latent construct accounts for >50% of the indicator variance [76]. We assessed internal consistency (reliability) using Werts, Linn, and Jöreskog's [77] composite reliability measure. Composite reliability is believed to be closer to actual reliability than Cronbach's alpha, which is a lower-bound estimate of reliability. Table 1 summarizes construct loading factors, Cronbach's alphas, squared AVE statistics, and composite reliabilities. The data indicate that the measures are robust regarding their internal consistency as indexed by the composite reliability score. The composite reliabilities of the variables exceed the recommended threshold value of 0.70 [78].

**Table 2. Baseline comparisons.**

| Model | NFI Delta1 | RFI rho1 | IFI Delta2 | TLI rho2 | CFI |
|---|---|---|---|---|---|
| Default model | .864 | .845 | .954 | .946 | .953 |
| Saturated model | 1.000 | | 1.000 | | 1.000 |
| Independence model | .000 | .000 | .000 | .000 | .000 |

The goodness-of-fit index (GFI) was used to measure the fit between the hypothesized model and the observed covariance matrix, GFI = 0.868. The model shows 276 distinct sample moments, with 55 distinct parameters to be estimated. The Chi-square totals 320.167 with 221 degrees of freedom, with a CMIN/DF of 1.449 and a probability level of 0.000. The difference between observed and expected covariance matrices is indicated by the Chi-squared test [79].

The model's reliability was endorsed using the root mean square error of approximation (RMSEA). The value of 0.0 supports the model as the literature considers the maximum to be 0.08 [80]. RMSEA of about .05 or less shows a close fit of the model about the degrees of freedom. The baseline comparisons fit indices suggest that the hypothesized model fits the observed variance-covariance matrix well compared to the null or independence model, as shown in Table 2. The Baseline Comparison indices are above the established limit of 0.7 [81].

## Results

### Confirmed hypotheses

The findings (Table 3 and Fig 4) show a solid and positive relationship between emotional intelligence (EI) and organizational citizenship behavior (OCB) (H1) ($b = 0.68$, $p<0.001$), showing that organizational citizenship behavior is driven by the emotional intelligence of the individuals in the selected sample and therefore confirming H1. This argument is supported by Yaghoubi [82], who states that emotional intelligence is significantly correlated with conscientiousness, civic virtue, and the altruistic behaviors of followers.

Hypothesis 7, which states that organizational citizenship behavior (OCB) positively impacts operational effectiveness (OE) ($b = 0.42$, $p<0.001$), was confirmed by the results. From the confirmation of hypotheses 1 and 7, we can infer that the impact of emotional intelligence on operational effectiveness is indirect through the mediation of organizational citizenship behavior. This is a new finding for the literature, as this result has not been explored before. There is no evidence in the literature of similar results to support this argument.

Furthermore, transactional leadership (TSL) has a predictive effect on operational effectiveness (OE) ($b = 0.27$, $p<0.001$), therefore, confirming hypothesis 9. This result indicates that

**Table 3. Regression weights: (Group number 1—Default model).**

| | | | Estimate | S.E. | C.R. | P | Label |
|---|---|---|---|---|---|---|---|
| OCB | <— | EI | .619 | .084 | 7.363 | *** | H1-Confirmed |
| TSL | <— | EI | .362 | .182 | 1.989 | .047 | H2-Partially confirmed |
| TFL | <— | EI | .032 | .115 | .277 | .782 | H3-Not confirmed |
| TSL | <— | OCB | .224 | .211 | 1.062 | .288 | H6-Not confirmed |
| TFL | <— | OCB | .083 | .135 | .615 | .539 | H5-Not confirmed |
| OE | <— | EI | .208 | .088 | 2.374 | .018 | H4-Partially confirmed |
| OE | <— | OCB | .391 | .108 | 3.601 | *** | H7-Confirmed |
| OE | <— | TFL | -.013 | .060 | -.212 | .832 | H8-Not confirmed |
| OE | <— | TSL | .163 | .047 | 3.436 | *** | H9-Confirmed |

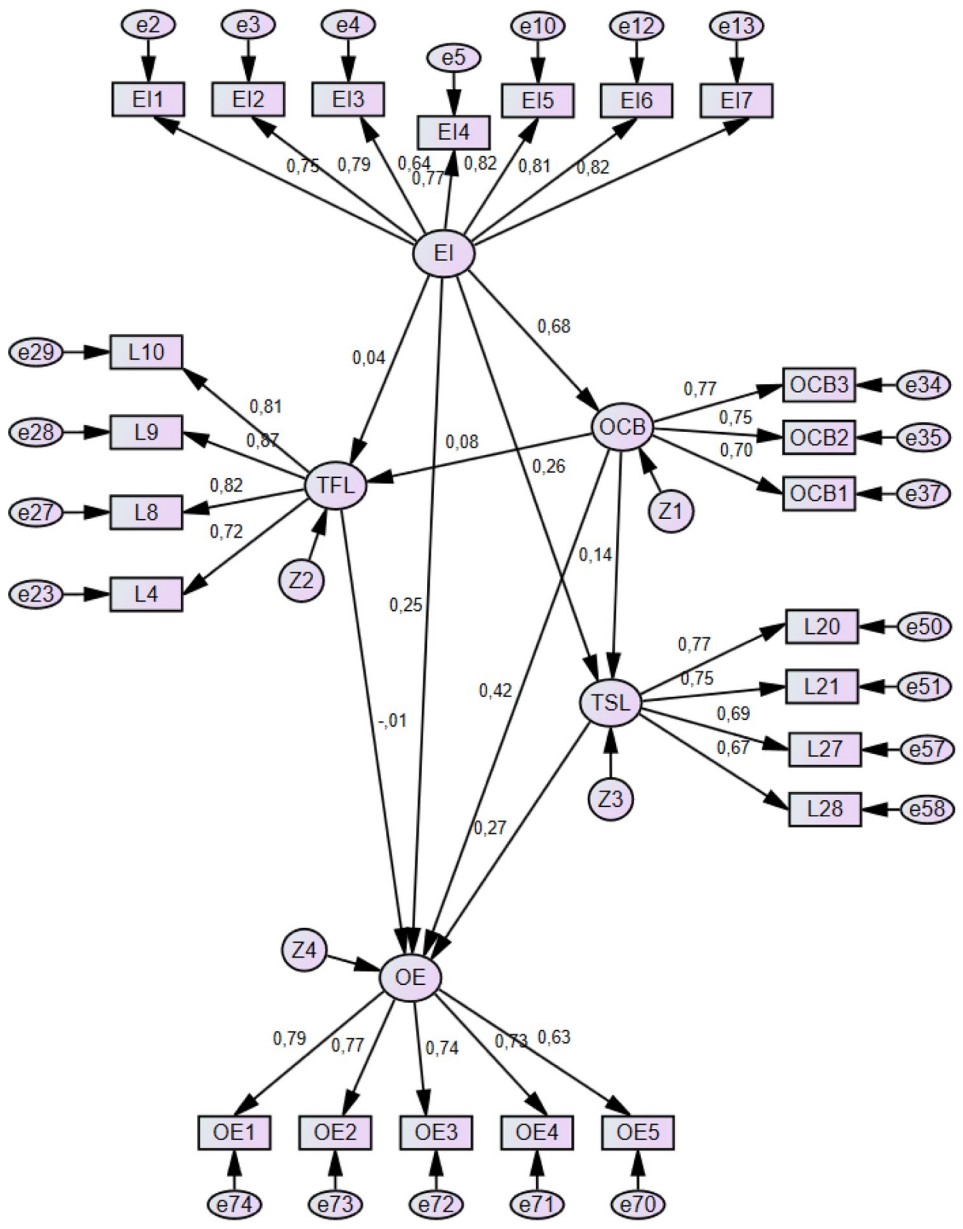

**Fig 4. Structural model.**

the leader acts as a change agent in the studied sample and significantly transforms subordinates to improve productivity. Since employees know what they desire, the leader explains what the subordinates will receive if their work aligns with the organization's operational effectiveness expectations. Additionally, the leader rewards the work of associates and is receptive to the personal interests of followers [37].

## Not confirmed hypotheses

In contrast, several hypotheses were not confirmed by the results, as is the case of hypothesis 3, which states that emotional intelligence (EI) has a positive impact on transformational

leadership (TFL) (*b = 0.04, p>0.05)*. For this to occur, leaders must strengthen their emotional intelligence before impacting and inspiring employees through transformational leadership. In the sample selected for this study, transformational leaders are not motivating their followers to perform above expectations.

Transformational leaders are charismatic individuals who engage their team members' intellect and emotions rather than dictating to them, inspiring and encouraging them instead of ordering them around, and kindling a positive vision inside their team. One characteristic of a transformational leader is the capacity to create an atmosphere where followers perform above and beyond expectations [39].

Hypothesis 5, which states that organizational citizenship behavior (OCB) positively impacts transformational leadership (TFL) (b = 0.08, *p>0.05*) was not confirmed by this study. The factors that motivate employees, such as altruism, conscientiousness, deportment, courtesy, and civic virtue, do not encourage transformational leadership.

Hypothesis 6, which postulates that there is a positive relationship between organizational citizenship behavior (OCB) and transactional leadership (TSL) (b = *0.14, p>0.05*), was not confirmed. This indicates that the main characteristics of organizational citizenship behavior are not predictors of transactional leadership, contradicting what the literature specifies [38, 83].

Hypothesis 8 suggests that transformational leadership (TFL) positively impacts operational effectiveness (OE) (b = -0.1, *p>0.05*). However, the results indicate that this is not the case in the studied sample, as hypothesis 8 was not confirmed. This outcome is because transformational leaders do not inspire employees to perform above the organization's expectations instead of their personal goals, indicating that leaders need to develop a better and more effective transformational leadership style. To achieve better transformational leadership skills, leaders could implement a motivation-based system that rewards those that meet their goals, providing positive reinforcement for a job well done—also known as a contingent reward [38].

## Partially confirmed hypotheses

Finally, this research obtained a third type of result: partially confirmed hypotheses, as is the case of hypothesis 2, where the results showed that transactional leadership (TSL) is partially impacted by emotional intelligence (EI) (b = 0.26, p<0.05). Transactional leadership is essentially a process in which leaders identify employees' emotional states, try to stimulate positive moods, and then attempt to regulate employees' emotions appropriately [84]. This result also denotes that leaders must reinforce the importance of transactional leadership practice.

Hypothesis 4, postulating that emotional intelligence (EI) positively impacts operational effectiveness (OE), also presents itself as a partially confirmed hypothesis (b = 0.25, *p<0.05*). This result demonstrates the importance of emotional intelligence in improving the organization's performance. For this study, the performance was measured through the achievement of operational effectiveness. Therefore, employees with higher emotional intelligence are more conscious of the importance of eliminating waste and achieving higher standards of quality, reliability, flexibility, and speed.

## Discussion

Organizations are established by individuals who can uphold links with their colleagues through emotional intelligence, competence, and skills. In essence, high emotional intelligence promotes employees' discretionary behavior in helping other members of the organization in their specific tasks or relevant organizational problems, which generates a feeling that coworkers help each other and go beyond the minimum required. Different behaviors, such as attendance to work, punctuality, and protection of resources to prevent dejection resulting from

inevitable difficulties, are encouraged by individuals with high emotional intelligence. Individuals with high emotional intelligence also promote work for the shared benefits of the organization through the appropriate management of operations.

Khalili [85] affirms that implementing transformational leadership toward employees with good emotional intelligence will increase their willingness to be involved in organizational citizenship behavior. However, our study indicates that individuals with high emotional intelligence also have high levels of organizational citizenship behavior, but these soft skills have no impact on transformational or transactional leadership. Although emotional intelligence partially or moderately affects transactional leadership, this is insufficient. Our study found that members of the chosen sample who exhibit high levels of organizational citizenship behavior and emotional intelligence do not support leaders in inspiring them to outperform above expectations, even if they are rewarded or punished for developing their tasks. Furthermore, our study does not support the argument by Podsakoff et al. [38], who pointed out that giving employees goals and aspirations and a suitable role model should encourage them to improve their performance. Leaders are not influencing the employees in the selected sample as their behavior is not based on contingent reward and punishment behavior, according to the argument stated by Podsakoff et al. [38].

This research backs the assertion made by Avolio et al. [86] that transactional leadership style impacts performance because H9 was verified. The attainment of operational effectiveness served as the benchmark for measuring performance in our study, and there is no correlation between transformational leadership and operational effectiveness, so H8 was not confirmed. Thus the results of this study only partially support the position of Purwanto et al. [83], who showed that the transactional and transformational leadership styles have a positive and strong and significant effect on performance. However, it is important to highlight that only transactional leadership impacts operational effectiveness, which supports the argument by Avolio et al. [86], who stated that reward behavior correlates positively with the subordinate's performance behavior.

## Conclusions

The literature demonstrates that leadership significantly affects employee performance [4]. However, few empirical studies show how leadership style and soft skills affect operational effectiveness. Consequently, this research explains the impact of emotional intelligence and organizational citizenship behavior on operational effectiveness when mediated by transformational and transactional leadership.

The findings of this study make a significant contribution to leadership and organizational behavior literature in the manufacturing sector and propose that organizations should implement practices that help enhance organizational citizenship behavior to achieve higher operational effectiveness and, consequently, a more sustainable competitive advantage.

The findings of this research evidence that organizational citizenship behavior has a more significant impact on operational effectiveness than does emotional intelligence. This result suggests that managers should reinforce soft skills such as organizational citizenship behavior in search of operational effectiveness and competitiveness. The study's results indicate that employees must develop emotional intelligence to carry out tasks more successfully. Still, only those with greater degrees of organizational citizenship behavior can significantly influence the efficiency and effectiveness of business operations. Additionally, this finding suggests that people with high emotional intelligence are more likely to engage in actions that promote cooperation among coworkers and better understand the feelings of others around them.

Additionally, this study contributes to the body of knowledge by demonstrating that emotional intelligence and organizational citizenship behavior are not strong predictors of either transactional or transformational leadership. In the chosen sample, managers are influenced by their staff members' civic engagement and often reward it. However, other research suggests that transformational leadership can boost organizational citizenship behavior, which is demonstrated by employees' willingness to go above and beyond what is required of them, take responsibility for their work, assist others in carrying out their duties, and pay attention to colleagues at work on a personal and professional level [87, 88]. However, this is not the case for the selected sample for this study.

Finally, it is essential to highlight that operational effectiveness is affected by organizational citizenship behavior and transactional leadership and partially by emotional intelligence in the selected sample. Therefore, this is an important lesson for managers and academics as it is necessary to explore strategies for transformational leadership to be more effective. The fact that employees see transactional leadership as a motivating style could be explained by the fact that the sample belongs to a developing economy with low salaries and suboptimal working conditions where employees expect to be rewarded for their behavior.

An organization will perform better and satisfy the demands for innovation and change from both the internal and external environment as a result of a transformational leadership style. A transformational leader is crucial for innovative practices because they value individual initiatives that can benefit the organization and the modifications brought about by the team members' creative methods [38–40].

## Further research

It is necessary to analyze voluntary and altruistic employee performance with other variables. For example, the negative perceptions of political behavior negatively influence employees' citizenship behavior. Political behavior is the informal, allegedly parochial, frequently controversial, and technically unlawful conduct not supported by institutional authority, accepted philosophy, or recognized competence. In this aspect, another dimension for studying organizational citizenship behavior could be the organizational climate. A significant positive relationship exists between the corporate environment and team performance mediated through psychological empowerment [89]. It would be interesting to know how political behavior and organizational climate relate to leadership and emotional intelligence.

## Limitations

There are a couple of limitations to this study. First, we used a convenience sample of respondents, carefully selecting them based on their role, knowledge, experience, ability, and tenure. Second, the sample size is small compared to more extensive quantitative studies conducted in other countries, so generalizability across sectors is not advised. Nonetheless, our findings supply insights that justify more comprehensive quantitative studies.

## Supporting information

**S1 Appendix. Measurement items.**
(DOCX)

## Author Contributions

**Conceptualization:** Ricardo Santa, Andreina Moros, Diego Morante, Annibal Scavarda.

**Formal analysis:** Ricardo Santa, Diego Morante.

**Investigation:** Ricardo Santa.

**Methodology:** Ricardo Santa, Diego Morante.

**Project administration:** Ricardo Santa.

**Supervision:** Ricardo Santa.

**Writing – original draft:** Ricardo Santa, Andreina Moros, Diego Morante.

**Writing – review & editing:** Ricardo Santa, Andreina Moros, Diego Morante, Dorys Rodríguez, Annibal Scavarda.

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
