## [Decision Letter · Decision Letter 0]

6 Mar 2023

PONE-D-23-04031The impact of emotional intelligence on operational effectiveness: The mediating role of organizational citizenship behavior and leadershipPLOS ONE

Dear Dr. Santa,

Thank you for submitting your manuscript to PLOS ONE. After careful consideration, we feel that it has merit but does not fully meet PLOS ONE’s publication criteria as it currently stands. Therefore, we invite you to submit a revised version of the manuscript that addresses the points raised during the review process.

We look forward to receiving your revised manuscript.

Kind regards,

Ahmad Samed Al-Adwan

Academic Editor

PLOS ONE

Journal Requirements:

"No authors have competing interest"

5. Please include a copy of Table 3 which you refer to in your text on page 14.

Additional Editor Comments:

After a thorough review of your manuscript, I would like to commend you on the novelty of your research topic and the insights that your study presents. Your paper addresses a critical issue in your field of study and has the potential to contribute significantly to the existing literature. However, your paper is currently is subjected to major revisions. The reviewers have pointed out several significant issues with your manuscript that need to be addressed before it can be considered for publication. The revisions required are substantial and will require a considerable amount of work on your part.

Reviewers' comments:

Reviewer's Responses to Questions

**Comments to the Author**

1. Is the manuscript technically sound, and do the data support the conclusions?

Reviewer #1: Yes

Reviewer #2: Partly

2. Has the statistical analysis been performed appropriately and rigorously? 

Reviewer #1: Yes

Reviewer #2: Yes

3. Have the authors made all data underlying the findings in their manuscript fully available?

Reviewer #1: Yes

Reviewer #2: Yes

4. Is the manuscript presented in an intelligible fashion and written in standard English?

Reviewer #1: Yes

Reviewer #2: Yes

5. Review Comments to the Author

Reviewer #1: Actually, this is a good paper with a robust research model . However, I have some concerns that the authors need fulfil.

- It is important to include a sub-section in the introduction section to highlight the research significance.

- Additionally, it is useful to highlight the structure of the paper that the end of the introduction section.

- The multi-collinearity, convergent validity, and discriminant validity tests are missed from the analysis section.

- What are the practical implications of this study? They seem lost in the conclusion section. Please report separately.

Reviewer #2: 1. The introductory section should conclude by emphasizing the significance of this study. Additionally, it is crucial to justify the originality of this paper by highlighting its principal contributions to the current body of literature.

2. Please provide a definition of the study population and the sampling technique utilized in the methodology and research design section. Additionally, please provide a rationale for why the selected sampling technique and sample size are suitable for the study.

3. The discussion should be strengthening by highlighting the novel findings. Furthermore, it is suggested to compare the current findings with previous well-established studies.

4. Both theoretical and practical contributions should be reported separately after the discussion section.

5. Finally, the hypothesis development section and discussion section can be supported by including relevant and recent studies. This encompasses, but is not restricted to:

• Organisational Culture and Organisational Citizenship Behaviour: The Dark Side of Organisational Politics. https://doi.org/10.2478/orga-2021-0003

• Organizational climate and team performance: the mediating role of psychological empowerment at Jordanian pharmaceutical companies. https://doi.org/10.1504/IJMP.2019.098661

• The Effects of Transformational Leadership, Organizational Innovation, Work Stressors, and Creativity on Employee Performance in SMEs. https://doi.org/10.3389/fpsyg.2022.772104

-

6. PLOS authors have the option to publish the peer review history of their article (what does this mean?). If published, this will include your full peer review and any attached files.

Reviewer #1: **Yes: **Husam Yaseen

Reviewer #2: **Yes: **Amro Al-Madadha

---

## [Author Response · Author response to Decision Letter 0]

1 Apr 2023

Answer to reviewers

Reviewer comment Action

Please ensure that your manuscript meets PLOS ONE's style requirements, including those for file naming. The PLOS ONE style templates can be found at We double-checked the Plos One format style, and we are sure our article is in the correct format 

Please provide additional details regarding participant consent. In the ethics statement in the Methods and online submission information, please ensure that you have specified what type you obtained (for instance, written or verbal, and if verbal, how it was documented and witnessed). If your study included minors, state whether you obtained consent from parents or guardians. If the need for consent was waived by the ethics committee, please include this information.

 In the document we included the following statement:

To avoid the social desirability bias, the ethics committee at the Colegio de Estudios Superiores de Administración (CESA) in Colombia reviewed the questionnaire to ensure that questions were neutral, unbiased, and non-threatening. Additionally, before filling out the questionnaire, the respondents were informed that all information provided would be treated in the strictest confidence, that the responses would be aggregated and used for research purposes, and that by completing this survey, they were giving their consent to such use, and that they might withdraw at any time. The researchers followed ethical procedures during this research project by adhering to ethical standards. 

Additionally, we included the following statement in the submission page in the ethical statement section:

The study was approved by the Ethics Committee at Colegio de Estudios Superiores de Administración (CESA). Approval number 006 date 25-04-2022

Committee Members: Dr. Edgardo Cayón (Research Director); Dr. Rodrigo Zarate (Full-time Professor); Dr. Javier Cadena (Full-time Professor).

Date: April 25th, 2022

The IRB Clearance document is attached under "Other" in this submission

The questionnaire respondents are given the following information before filling out the questionnaire:

CONFIDENTIALITY ASSURANCE

• All information provided will be treated in strictest confidence. 

• The responses will be aggregated and used to examine research issues. 

• At no stage will any of the information be divulged to third parties in their disaggregated form.

• You may withdraw at any time.

• By completing this survey you are giving your consent.

Ethical procedures were adopted by the researchers when building this research project.

Thank you for stating the following in your Competing Interests section: 

"No authors have competing interest"

 This information was provided accordingly

Please include your full ethics statement in the 'Methods' section of your manuscript file. In your statement, please include the full name of the IRB or ethics committee who approved or waived your study, as well as whether or not you obtained informed written or verbal consent. If consent was waived for your study, please include this information in your statement as well. 

 A full ethics statement was provided in the Methodology section 

Please include a copy of Table 3 which you refer to in your text on page 14. This was an error in table numbering and has now been corrected.

Reviewer #1: Actually, this is a good paper with a robust research model . However, I have some concerns that the authors need fulfil.

- It is important to include a sub-section in the introduction section to highlight the research significance.

 We addressed the originality value in the abstract as follows:

The value of this paper lies in discerning the current capabilities and strategies that individuals in an organization must address for proper transactional and transformational leadership. However, before operational effectiveness and a sustainable competitive advantage can be achieved, the role of leaders should be managed through the appropriate application of the concepts of emotional intelligence and organizational leadership behavior.

- Additionally, it is useful to highlight the structure of the paper that the end of the introduction section.

 The article has a paragraph at the end of the introduction describing the structure of the paper:

This article is organized as follows. The second section provides the literature review supporting this study's model. The third section presents the hypothetical model and the methods of empirical analysis. The fourth section offers the quantitative results, and the fifth section discusses these results and provides the analysis and implications of the findings. The last section concludes the investigation. 

- The multi-collinearity, convergent validity, and discriminant validity tests are missed from the analysis section.

 The issue of multicollinearity, convergent validity, and discriminant validity tests was resolved with the introduction of Table 1. Reliability measures in addition to the following explanation:

The state of multicollinearity among dimensions was examined using AMOS. There was no evidence of multicollinearity in the data. The variance inflation factor (VIF), derived from the regression analysis, was used to determine how much one independent variable interacted with another. A typical assessment of multicollinearity is the VIF value.

To avoid overfitting, we considered the fitting propensity of the model through the evaluation of the parsimony indices, PNFI=0.755, PCFI=0.832, and PGFI=0.695 [75].

We compared the square root of the average variance extracted (AVE) to test the model's reliability. AVE measures the amount of indicator variance explained by the latent variable relative to the amount due to measurement error [76]. Ideally, the AVE should be >0.50. This means that the latent construct accounts for >50% of the indicator variance [77]. We assessed internal consistency (reliability) using Werts, Linn, and Jöreskog's [78] composite reliability measure. Composite reliability is believed to be closer to actual reliability than Cronbach's alpha, which is a lower-bound estimate of reliability. Table 1 summarizes construct loading factors, Cronbach’s alphas, squared AVE statistics, and composite reliabilities. The data indicate that the measures are robust regarding their internal consistency as indexed by the composite reliability score. The composite reliabilities of the variables exceed the recommended threshold value of 0.70 [79].

- What are the practical implications of this study? They seem lost in the conclusion section. Please report separately. The discussion and conclusion have been rewritten to address the issue raised by the reviewer. Additions to the conclusion include the following:

The literature demonstrates that leadership significantly affects employee performance [4]. However, few empirical studies show how leadership style and soft skills affect operational effectiveness. Consequently, this research explains the impact of emotional intelligence and organizational citizenship behavior on operational effectiveness when mediated by transformational and transactional leadership. 

The findings of this study make a significant contribution to leadership and organizational behavior literature in the manufacturing sector and propose that organizations should implement practices that help enhance organizational citizenship behavior to achieve higher operational effectiveness and, consequently, a more sustainable competitive advantage. 

The findings of this research evidence that organizational citizenship behavior has a more significant impact on operational effectiveness than does emotional intelligence. This result suggests that managers should reinforce soft skills such as organizational citizenship behavior in search of operational effectiveness and competitiveness. The study's results indicate that employees must develop emotional intelligence to carry out tasks more successfully. Still, only those with greater degrees of organizational citizenship behavior can significantly influence the efficiency and effectiveness of business operations. Additionally, this finding suggests that people with high emotional intelligence are more likely to engage in actions that promote cooperation among coworkers and better understand the feelings of others around them. 

Additionally, this study contributes to the body of knowledge by demonstrating that emotional intelligence and organizational citizenship behavior are not strong predictors of either transactional or transformational leadership. In the chosen sample, managers are influenced by their staff members' civic engagement and often reward it. However, other research suggests that transformational leadership can boost organizational citizenship behavior, which is demonstrated by employees' willingness to go above and beyond what is required of them, take responsibility for their work, assist others in carrying out their duties, and pay attention to colleagues at work on a personal and professional level [88, 89]. However, this is not the case for the selected sample for this study. 

Finally, it is essential to highlight that operational effectiveness is affected by organizational citizenship behavior and transactional leadership and partially by emotional intelligence in the selected sample. Therefore, this is an important lesson for managers and academics as it is necessary to explore strategies for transformational leadership to be more effective. The fact that employees see transactional leadership as a motivating style could be explained by the fact that the sample belongs to a developing economy with low salaries and suboptimal working conditions where employees expect to be rewarded for their behavior. 

An organization will perform better and satisfy the demands for innovation and change from both the internal and external environment as a result of a transformational leadership style. A transformational leader is crucial for innovative practices because they value individual initiatives that can benefit the organization and the modifications brought about by the team members' creative methods [38-40].

Reviewer #2: 1. The introductory section should conclude by emphasizing the significance of this study. Additionally, it is crucial to justify the originality of this paper by highlighting its principal contributions to the current body of literature.

 The introduction section clearly highlights the contribution to the body of the literature:

Organizations in the manufacturing sector seek operational effectiveness by reducing costs and improving the quality of their processes [5, 6]. Employees' skills and knowledge can be crucial assets for this purpose, regardless of the operational level of the organization's position in the market [7]. Studies in service sector organizations have empirically shown that emotional intelligence and organizational citizenship behavior significantly impact an organization's competitiveness [8]. Others have concluded that leadership style positively influences organizational performance and customer satisfaction [9]. Therefore, it is crucial to investigate the role of employees' soft skills in achieving sustainable competitive advantage. Additionally, it is essential to understand better the impact of emotional intelligence and organizational citizenship behavior on operational effectiveness when mediated by the leadership style exercised by employees.

2. Please provide a definition of the study population and the sampling technique utilized in the methodology and research design section. 

 We added the following paragraph in the methodology section:

leadership responsibilities. Each received questionnaire was reviewed for completeness; several were considered unusable due to inconsistencies or significant missing data, or lack of involvement of the respondent in leadership responsibilities. In total, 76 questionnaires were discarded. The response rate was 59.30%. 

Additionally, we wrote the following clarification in the Limitations section:

There are a couple of limitations to this study. First, we used a convenience sample of respondents, carefully selecting them based on their role, knowledge, experience, ability, and tenure. Second, the sample size is small compared to more extensive quantitative studies conducted in other countries, so generalizability across sectors is not advised. Nonetheless, our findings supply insights that justify more comprehensive quantitative studies.

Additionally, please provide a rationale for why the selected sampling technique and sample size are suitable for the study. 

Please see the previous answer for the rationale.

With regard to the sample size, we added the following sentence:

The indications of Hair et. al [68] were followed to support the sample size used in this research. The minimum sample for an SEM model must be larger than the number of covariances in the data matrix input. Additionally, the minimum appropriate sample is made up of 10 respondents per parameter. Therefore 180 valid responses are more than sufficient for an SEM model with five constructs.

3. The discussion should be strengthening by highlighting the novel findings. Furthermore, it is suggested to compare the current findings with previous well-established studies.

 We followed the reviewers' advice, highlighting the novel findings and comparing them to current findings.

For example:

Discussion 

Organizations are established by individuals who can uphold links with their colleagues through emotional intelligence, competence, and skills. In essence, high emotional intelligence promotes employees' discretionary behavior in helping other members of the organization in their specific tasks or relevant organizational problems, which generates a feeling that coworkers help each other and go beyond the minimum required. Different behaviors, such as attendance to work, punctuality, and protection of resources to prevent dejection resulting from inevitable difficulties, are encouraged by individuals with high emotional intelligence. Individuals with high emotional intelligence also promote work for the shared benefits of the organization through the appropriate management of operations. 

Khalili [86] affirms that implementing transformational leadership toward employees with good emotional intelligence will increase their willingness to be involved in organizational citizenship behavior. However, our study indicates that individuals with high emotional intelligence also have high levels of organizational citizenship behavior, but these soft skills have no impact on transformational or transactional leadership. Although emotional intelligence partially or moderately affects transactional leadership, this is insufficient. Our study found that members of the chosen sample who exhibit high levels of organizational citizenship behavior and emotional intelligence do not support leaders in inspiring them to outperform above expectations, even if they are rewarded or punished for developing their tasks. Furthermore, our study does not support the argument by Podsakoff et. al [38], who pointed out that giving employees goals and aspirations and a suitable role model should encourage them to improve their performance. Leaders are not influencing the employees in the selected sample as their behavior is not based on contingent reward and punishment behavior, according to the argument stated by Podsakoff et. al [38]. 

This research backs the assertion made by Avolio et. al [87] that transactional leadership style impacts performance because H9 was verified. The attainment of operational effectiveness served as the benchmark for measuring performance in our study, and there is no correlation between transformational leadership and operational effectiveness, so H8 was not confirmed. Thus the results of this study only partially support the position of Purwanto et. al [84], who showed that the transactional and transformational leadership styles have a positive and strong and significant effect on performance. However, it is important to highlight that only transactional leadership impacts operational effectiveness, which supports the argument by Avolio et. al [87], who stated that reward behavior correlates positively with the subordinate's performance behavior.

4. Both theoretical and practical contributions should be reported separately after the discussion section.

 We rewrote the discussion and conclusion sections to address this issue. Please check the previous answer which clarifies this comment by the reviewer.

5. Finally, the hypothesis development section and discussion section can be supported by including relevant and recent studies. This encompasses, but is not restricted to:

• Organisational Culture and Organisational Citizenship Behaviour: The Dark Side of Organisational Politics. https://doi.org/10.2478/orga-2021-0003

• Organizational climate and team performance: the mediating role of psychological empowerment at Jordanian pharmaceutical companies. https://doi.org/10.1504/IJMP.2019.098661

• The Effects of Transformational Leadership, Organizational Innovation, Work Stressors, and Creativity on Employee Performance in SMEs. https://doi.org/10.3389/fpsyg.2022.772104

 We found the suggested articles very appropriate to enrich the discussion and support for the literature presented in this research. Therefore, the three references were included in the article.

---

## [Editor Report · Decision Letter 1]

10 Apr 2023

The impact of emotional intelligence on operational effectiveness: The mediating role of organizational citizenship behavior and leadership

PONE-D-23-04031R1

Dear Dr. Santa,

We’re pleased to inform you that your manuscript has been judged scientifically suitable for publication and will be formally accepted for publication once it meets all outstanding technical requirements.

Kind regards,

Ahmad Samed Al-Adwan

Academic Editor

PLOS ONE

Additional Editor Comments (optional):

Dear Editor,

I would like to thank you for resubmitting the revised version of your paper. The quality of this paper has enhanced significantly after addressing the reviewers' comments. Thus, I am satisfied with the current version.

Regards,
---

## [Editor Report · Acceptance letter]

20 Apr 2023

PONE-D-23-04031R1 

The impact of emotional intelligence on operational effectiveness: The mediating role of organizational citizenship behavior and leadership 

Dear Dr. Santa:

I'm pleased to inform you that your manuscript has been deemed suitable for publication in PLOS ONE. Congratulations! Your manuscript is now with our production department. 

Kind regards, 

on behalf of

Prof. Ahmad Samed Al-Adwan 

Academic Editor

PLOS ONE